

# Effectiveness of therapeutic ultrasound on reducing pain intensity and functional disability in patients with plantar fasciitis: a systematic review of randomised controlled trials

Anas Mohammed Alhakami[1], Reem Abdullah Babkair[1], Ahmad Sahely[2,3] and Shibili Nuhmani[1]

[1] Department of Physical Therapy, Imam Abdulrahman Bin Faisal University, Dammam, Saudi Arabia
[2] Physiotherapy Department, Faculty of Applied Medical Science, Jazan University, Jazan, Saudi Arabia
[3] School of Sports, Exercise and Rehabilitation Sciences, University of Birmingham, Birmingham, United Kingdom

Corresponding author
Shibili Nuhmani,
snuhmani@iau.edu.sa

## ABSTRACT

**Background:** Ultrasound therapy is one of the preferred conservative treatments for patients with plantar fasciitis. This study aims to evaluate the effectiveness of therapeutic ultrasound in decreasing pain intensity and improving functional disability in patients with plantar fasciitis.

**Methods:** Five randomised control trials (RCT) were selected based on an electronic search in PubMed, Trip Database and PEDro. To be included in the systematic review, the study should be an RCT which investigated the effectiveness of therapeutic ultrasound conducted in patients with plantar fasciitis with pain intensity and functional disability as outcome measures. Only studies published in peer-reviewed journals written in the English language were included. The quality of the selected studies was measured by the PEDro scale.

**Results:** All the included studies showed that ultrasound therapy is beneficial in reducing pain score and improving functional disability, except one study did not recommend using ultrasound therapy for plantar fasciitis. Moreover, regarding another outcome measure, two studies found that ultrasound therapy reduces thickness and tenderness in plantar fasciitis and improves static and dynamic balance.

**Conclusion:** After reviewing the five studies, this systematic review support using ultrasound therapy to decrease pain and improve functional disability in patients with plantar fasciitis.

**Study Registration:** https://osf.io/xftzy/.

## INTRODUCTION

Plantar fasciitis is a common and debilitating musculoskeletal condition, affecting approximately 10% of the general population during their lifetime (*Thomas et al., 2010*; *Trojian & Tucker, 2019*). Plantar fasciitis has many other names, including painful heel syndrome, heel spurs, runner's heels, sub-calcaneal discomfort, calcaneodynia, and calcaneal periostitis (*Gautham, Nuhmani & Kachanathu, 2014*; *Latt et al., 2020*). It develops when the plantar fascia gets inflamed, causing heel discomfort and pain (*Chhabra & Singh, 2021*; *Nuhmani, 2012*). It is a degenerative disorder of the plantar fascia that causes it to become overly thick and painful, particularly where it connects to the calcaneal bone. Because of the impacts of this disorder, 11–15% of those afflicted require medical care (*League, 2008*). This condition commonly affects runners and military personnel, although it may affect any population, especially middle-aged ladies between the ages of 40 and 60 (*Buchbinder, 2004*; *Pohl, Hamill & Davis, 2009*; *Rabadi et al., 2022*).

Abnormal intra-articular ankle loading, reduced ankle range of motion, especially in the dorsiflexion, overuse, obesity, and Achilles tendon injuries are the most common risk factors for this condition (*Tahririan et al., 2012*). Various symptoms characterise plantar fasciitis, the most prevalent of which is increased discomfort with initial steps in the morning and pain with continuous weight bearing (*Gautham, Nuhmani & Kachanathu, 2014*). Plantar fasciitis treatment may involve non-pharmacological treatment (conservative treatment) and pharmacological treatment, including non-steroidal anti-inflammatory drugs (NSAIDs) and opioids such as steroid injections and surgery (*Latt et al., 2020*; *Rhim et al., 2021*). Conservative management consists of electrotherapeutic modalities, which include Ultrasound, TENS, shockwave therapy, manual therapy, including joint and soft tissue manipulation; therapeutic exercises, which may include stretching and strengthening exercises; and other modalities, which may include taping, night splints, ice, heat, and orthosis (*Agostini et al., 2022*; *Gautham, Nuhmani & Kachanathu, 2014*; *Rhim et al., 2021*). It has been demonstrated that conservative treatment may alleviate 95% of symptoms in 12 to 18 months. Surgery may be considered if conservative treatment and steroid injection are ineffective after 6 months (*Nayar, Alcock & Vemulapalli, 2023*; *Thomas et al., 2010*).

Therapeutic ultrasound is the preferred choice of treatment in plantar fasciitis due to its duel benefit, first through the thermodynamic effect caused by the increased temperature of the soft tissue between 41 to 44 degrees, which helps to relax muscles, shorten the time it takes for muscles to contract, and increase the pliability of collagen fibers (*Baker, Robertson & Duck, 2001*; *Papadopoulos & Mani, 2020*). The second effect is an acoustic micro-streaming and cavitation-generated non-thermal effect (pulsed mode) that improves cell permeability and vascular circulation, thereby providing pain relief, and it may be used for both acute and subacute conditions (*Baker, Robertson & Duck, 2001*; *Papadopoulos & Mani, 2020*).

Several studies have reported positive effects of therapeutic ultrasound in people with plantar fasciitis. For instance, a study published in 2018 found that using therapeutic ultrasound reduced pain intensity and disability (*Akinoğlu et al., 2017*). Moreover, a study

by *Dedes et al. (2019)* reported that therapeutic ultrasound effectively reduced pain and improved function in people with plantar heel pain. On the other hand, some studies have failed to find significant effects of therapeutic ultrasound. A study comparing the effectiveness of low-level laser and ultrasound therapy found that therapeutic ultrasound was not more effective than low-level laser in reducing pain and improving function in people with plantar fasciitis (*Jothi Prasanna, Sherpa & Sivakumar, 2017*). Furthermore, an experimental study found no significant difference between active and placebo ultrasound in reducing pain and disability (*Crawford & Snaith, 1996*).

Despite several RCTs and case studies, ultrasound therapy's effectiveness in managing plantar fasciitis remains unclear. Therefore, this literature review aims to clarify the efficacy of therapeutic ultrasound in treating plantar fasciitis. The findings of this study will be helpful for the clinician in deciding whether to use ultrasound therapy for the management of plantar fasciitis.

## METHODS

*Research Question:* Does ultrasound help people with plantar fasciitis have less severe pain and a greater improvement in their capacity to function?

The research question was based on the PICO structure (Table 1) (Cochrane Handbook, Sec. 5.1.1; https://training.cochrane.org/handbook/current).

### Eligibility criteria

#### Inclusion criteria

The randomised controlled trials (RCTs) involving patients with plantar fasciitis over 18 years of age using therapeutic ultrasound as a treatment modality. Only English language studies that were published between January 1997 and October 2022 and used pain and functional disability as outcomes were included.

#### Exclusion criteria

Studies involving patients with peripheral neuropathy, Morton neuroma, diabetic foot, cancer, infection, Calcaneal fractures or involving participants who used steroids in the past 6 months or those who did foot surgery were excluded.

### Search strategy

The data searching included the following electronic databases: PubMed, Trip Database and PEDro. The keywords used for the search included: Plantar fasciitis, Plantar fasciitis, plantar faciopathy, joggers' heel, heel spur syndrome, ultrasound, therapeutic ultrasound or therapeutic, pain and functional disability. The Boolean "AND" operator was used to join the Subjects, whereas the "OR" operator was used to connect the headers. The search was conducted independently by two investigators on October 21, 2022. The data was compiled after the papers were validated based on their titles and abstracts. Studies discovered throughout the search led to further manuscripts *via* their cited works. Two independent reviewers (AMA & AS) evaluated the gathered studies separately, and the discrepancies were resolved by consensus.

| Table 1 Research question. | |
|---|---|
| P | Patients diagnosed with plantar fasciitis |
| I | Therapeutic ultrasound |
| C | Any other modalities or exercises other than ultrasound therapy |
| O | Pain intensity and functional disability |
| S | Study design-systematic review of randomised controlled trials |

## Study selection

The identified studies from the database search were transferred and saved into Endnote software. After removing the duplicates, the initial screening of the studies' titles was performed. The screening of abstracts was then done independently by reviewers (AMA & AS). Irrelevant studies were excluded, and the full-text screening was then done independently by the two reviewers. Conflicts between the two reviewers were resolved by a third reviewer (SN).

## Critical appraisal for methodological quality

The PEDro scale, which consists of eleven different components, was used to evaluate the quality of each selected study. The PEDro scale may grant 10 points, with zero being the lowest possible score and 10 being the most. The scale awards one point for each completed item, except for the first item. According to *de Morton (2009)*, the PEDro scale is a valid measurement scale, and the reliability of this scale varies from good to exceptional. On this scale, a study was rated as having excellent quality if it scored six or above, moderate or fair quality if it received a score between four and five, and poor quality if it received a score of three or less.

## Data extraction and synthesis

The primary investigator extracted the data, which were assembled into a special data extraction form. Title, author, year of publication, number of individuals enrolled, diagnosis, type of intervention (including intensity of intervention, frequency of intervention, number of total sessions, and number of sessions per week), outcome measures, main results and conclusion were retrieved from each research and summarised in tables.

# RESULTS

## Selection of studies

A total of 220 studies were found from the databases searched. After screening, four publications were found relevant to the scope of the review (*Akinoğlu et al., 2017*; *Katzap et al., 2018*; *Konjen, Napnark & Janchai, 2015*; *Ulusoy, Cerrahoglu & Orguc, 2017*), and one more study was selected through a manual search (*Greve, Grecco & Santos-Silva, 2009*), making a total of five studies. The PRISMA flow diagram for choosing and filtering the articles used in the study is shown in Fig. 1.

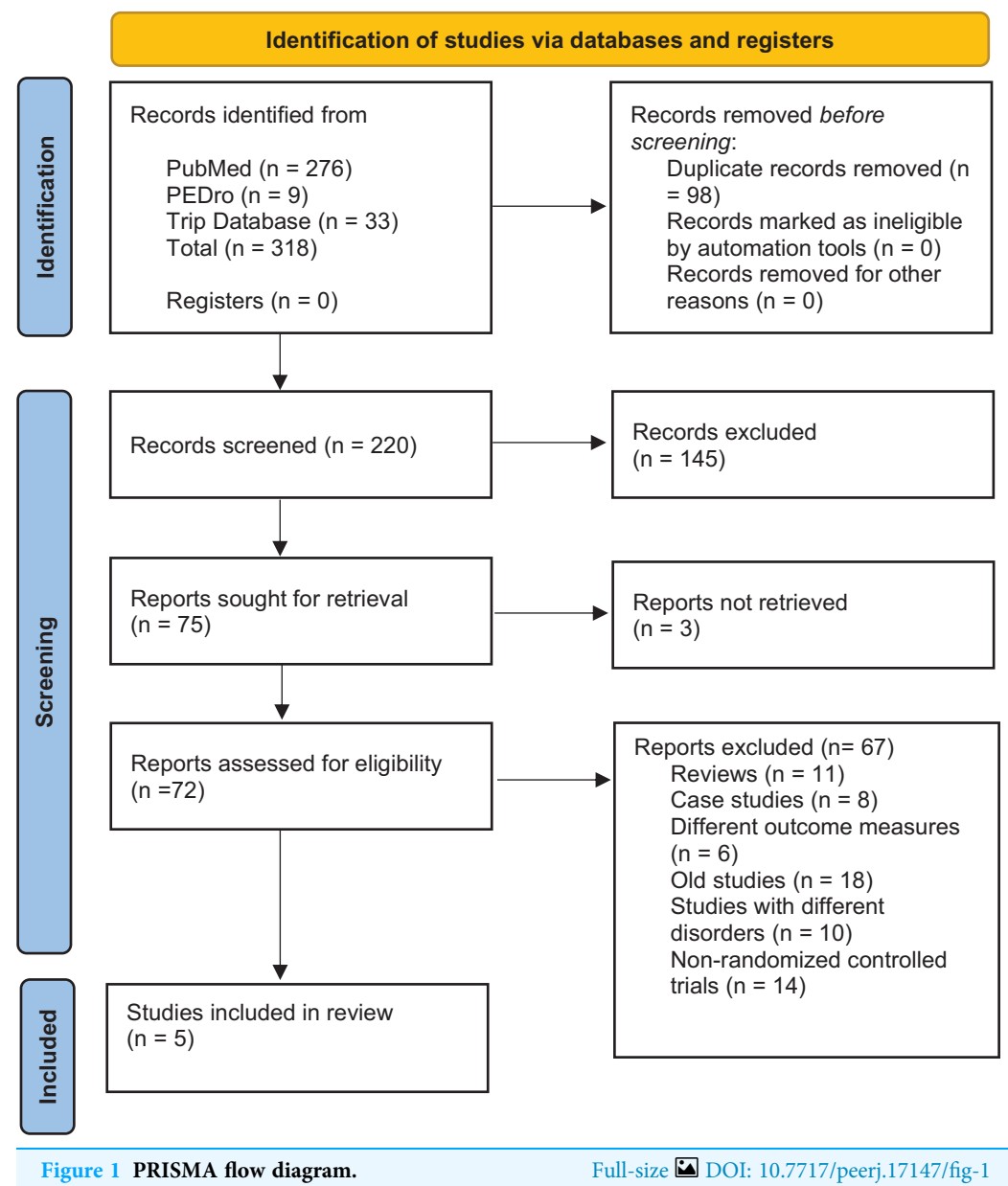

**Figure 1  PRISMA flow diagram.**

## Study characteristics

The characteristics of the selected studies are summarised in Table 2. The five selected studies recruited 207 participants. Four studies compared therapeutic ultrasound to shock wave therapy (*Akinoğlu et al., 2017*; *Greve, Grecco & Santos-Silva, 2009*; *Konjen, Napnark & Janchai, 2015*; *Ulusoy, Cerrahoglu & Orguc, 2017*), and one compared the therapeutic ultrasound plus stretching exercises to sham therapeutic ultrasound with stretching exercises (*Katzap et al., 2018*). Four studies used pain intensity and functional disability, such as the Numeric Pain Rating Scale (NPRS) (*Katzap et al., 2018*), the Visual Analog Scale (VAS) (*Akinoğlu et al., 2017*; *Greve, Grecco & Santos-Silva, 2009*; *Konjen, Napnark & Janchai, 2015*; *Ulusoy, Cerrahoglu & Orguc, 2017*) and functional disability such as the

**Table 2 Characteristics of the selected studies.**

| Authors | Participants | Duration of the symptoms | Diagnostic criteria | Interventions | Outcomes measures | Time of pain measurement | Results | Mean difference in results | Conclusion |
|---|---|---|---|---|---|---|---|---|---|
| *Katzap et al.* (2018) | 54 subjects (aged between 24 and 80 years. Intervention group = 28 and control group = 26 | Not mentioned | Gradual pain, Pian generated by carrying weight/ local pressure, Increase in pain in the morning upon first few steps or after prolonged non-weight-bearing, Symptoms decrease after slight activities | Ultrasound was used on the intervention group, while stretching exercises and sham ultrasound therapy were administered to the control group. | NPRS CAT | -First morning steps. -Middle of the day | Both groups showed improvement across the board when looking at the outcomes (active US and sham US). After the therapy was completed, no statistically significant difference existed between the groups in any of the outcome measures. | -NPRS- morning: 01 (−1.07, 1.09). -NPRS-mid day: 58 (−0.42, 1.58). -CAT: 44 (−3.61, 6.49). | The inclusion of the US in the treatment plan had little effect. As a result, the researchers concluded that ultrasound should not be used to treat plantar fasciitis. |
| *Ulusoy, Cerrahoglu & Orguc* (2017) | Sixty subjects (aged between 19 and 75 years). Group 1 = 20, group 2 = 20 and group 3 = 20 | More than 6 months | Plantar pain Tenderness to palpation with local pressure at plantar fascia origin, Pain in the first few steps in the morning/during daily activities/ during exercise | LLTL was used in Group 1, ultrasound was used in Group 2, and electrostimulation was used in Group 3. | VAS AOFAS HTI MRI | -First morning steps. -During daily activities. -During exercises. | The VAS, AOFAS, and HTI indicated improvement across all three groups. In addition to this, the MRI indicated a reduction in thickness across all of the groups. | Not mentioned | In LLLT, the treatment success rate was 60.6%, whereas in ESWT, it was 65%, and in the US, it was 23.5%. The effectiveness of ESWT and LLLT was superior to that of the US. |
| *Akinoglu et al.* (2017) | Fifty-four subjects (aged between 32 and 65 years). Groups 1 and 2 each had 18 subjects and 18 control group participants. | More than 3 months | Pain on palpation, Heel spur in lateral radiograph | Intervention group 1 was treated with r-ESWT (weekly once for 3 weeks) and home exercises (twice daily for 4 weeks). Intervention group 2 was treated with ultrasound (twice weekly for seven sessions) and home exercises (twice daily for 4 weeks). The only treatment that was given to the control group was home exercises (twice a day for 4 weeks) | FFI AOFAS Single-leg stance test Functional reach test Ankle proprioception | Not mentioned | The FFI score decreased across the board, although the change was more pronounced in the US group (*P* 0.05). All groups showed an increase in AOFAS. However, the rise in the control group was significantly less (*P* 0.05). Static and dynamic balance improved with a *P* value less than 0.05. However, only the r-ESWT group saw an improvement in their perception of proprioception in the ankle (*P* 0.05). | Not mentioned | All of the groups, but especially the r-ESWT and US groups, saw an improvement in their symptoms. On the other hand, the FFI score dropped more in the US group, and the r-ESWT group was the only one that improved proprioception. |

| Authors | Participants | Duration of the symptoms | Diagnostic criteria | Interventions | Outcomes measures | Time of pain measurement | Results | Mean difference in results | Conclusion |
|---|---|---|---|---|---|---|---|---|---|
| Konjen, Napnark & Janchai (2015) | Thirty subjects (ages between 43 and 47 years). Intervention group1 = 15 and intervention group2 = 15 | More than 3 months | Heel pain in the first few steps in the morning (VAS more than 5) | Intervention group 1 received r-ESWT treatment (once a week for a total of six sessions), while intervention group 2 received ultrasound (US) treatment (3 sessions per week for 18 sessions) | VAS PFPS Treatment satisfaction | First morning steps. | When patients in both the r-ESWT and US groups were examined following treatment at 1, 3, 6, 12, and 24 weeks, there was a significant reduction in pain and increased mobility ($P$ 0.001). The r-ESWT group reported greater levels of patient satisfaction. | Not mentioned | Both r-ESWT and US were beneficial in reducing pain and enhancing patient movement. However, the r-ESWT performed far better than the US. |
| Greve, Grecco & Santos-Silva (2009) | Thirty-two subjects (aged between 20 and 68 years). Intervention group 1 = 16 and intervention group 2 = 16 | Not mentioned | Ultra sonography—planatr fascia more than 4mm thickness | The first intervention group had treatment with ultrasound (twice a week for a total of 10 sessions), in addition to kinesiotherapy and stretching exercises. In the second intervention, patients were given stretching exercises and radial shockwave treatment once a week for three sessions. | VAS Periodicity of pain Duration of pain Fincher's algometer | -Morning pain -During gait -Orthostatic position pain | The patient had less pain and improved function due to both therapies. | Not mentioned | When patients were re-examined 3 months after receiving shockwave therapy, it was determined that the treatment was not any more successful than in the US. |

**Note:**
NPRS, Numeric pain rating scale; CAT, computerised adaptive test; US, therapeutic ultra sound; VAS, visual analogue scale; AOFAS, American orthopaedic foot and ankle society; HTI, heel tenderness index; LLLT, low-level laser therapy; ESWT, extracorporeal shock wave therapy.

**Table 3 Critical appraisal of the selected studies.** PEDro scale score.

| Study | 1* | 2 | 3 | 4 | 5 | 6 | 7 | 8 | 9 | 10 | 11 | Score |
|---|---|---|---|---|---|---|---|---|---|---|---|---|
| *Katzap et al. (2018)* | ✓ | ✓ | ✓ | ✓ | ✓ | X | ✓ | ✓ | X | ✓ | ✓ | 8/10 |
| *Ulusoy, Cerrahoglu & Orguc (2017)* | ✓ | ✓ | X | ✓ | X | X | ✓ | ✓ | X | ✓ | ✓ | 6/10 |
| *Akinoğlu et al. (2017)* | ✓ | ✓ | ✓ | ✓ | X | X | X | X | X | ✓ | ✓ | 5/10 |
| *Konjen, Napnark & Janchai (2015)* | ✓ | ✓ | ✓ | ✓ | X | X | X | ✓ | X | ✓ | ✓ | 6/10 |
| *Greve, Grecco & Santos-Silva (2009)* | X | ✓ | X | ✓ | X | X | X | ✓ | X | ✓ | ✓ | 5/10 |

Notes:
1*, Criteria for eligibility have been outlined.
2, The use of a randomiser.
3, Hidden allotment of resources.
4, The groups were comparable before we started.
5, The participants will be blinded.
6, A blinding of the therapists.
7, Assessors blinding.
8, Adequate subsequent action.
9, An examination of the intention to treat.
10, Comparisons based on statistics between the groups.
11, Point measurements and a measure of the variability of the data.

American Orthopaedic Foot and Ankle Society (AOFAS) hind foot score (*Akinoğlu et al., 2017*; *Ulusoy, Cerrahoglu & Orguc, 2017*), the Plantar Fasciitis Pain and Disability Scale (PFPS) (*Konjen, Napnark & Janchai, 2015*), the Foot Function Index (FFI) (*Akinoğlu et al., 2017*) and the Computerized Adaptive Test (CAT) (*Katzap et al., 2018*) as outcome measures and one study measured the pain intensity only (VAS) (*Greve, Grecco & Santos-Silva, 2009*). In addition, some other outcome measures used across the reviewed studies such as; one study used pain periodicity and pain duration (*Greve, Grecco & Santos-Silva, 2009*), one study used ankle proprioception sense and static equilibrium (*Akinoğlu et al., 2017*), two studies used pressure pain threshold (*Greve, Grecco & Santos-Silva, 2009*; *Katzap et al., 2018*), one study used tenderness (*Ulusoy, Cerrahoglu & Orguc, 2017*) and plantar fascia thickness (*Ulusoy, Cerrahoglu & Orguc, 2017*). Out of the five selected studies, only three reported the duration of the symptoms. The participants in the *Ulusoy, Cerrahoglu & Orguc (2017)* study had plantar heel pain for more than 6 months, whereas in the other two studies (*Akinoğlu et al., 2017*; *Konjen, Napnark & Janchai, 2015*), the duration of the symptoms were more than 3 months. The diagnostic/inclusion criteria were also varied between the selected studies (Table 2). Only one study (*Greve, Grecco & Santos-Silva, 2009*) used ultrasonography for the diagnostic purpose of the condition. The most common inclusion criteria which were used in four studies was 'pain in the first few steps in the morning'.

## Methodological quality

The methodological quality of the selected studies assessed by the PEDro scale is available in Table 3. The score ranged between 5 and 8. Three studies were considered high quality (*Katzap et al., 2018*; *Konjen, Napnark & Janchai, 2015*; *Ulusoy, Cerrahoglu & Orguc, 2017*), scoring between 6 and 8 on the scale. Two studies scored five and were rated as fair/moderate quality (*Akinoğlu et al., 2017*; *Greve, Grecco & Santos-Silva, 2009*).

### Effect of ultrasound therapy on pain intensity

Four of the selected five studies showed that US therapy relieved the pain intensity (*Akinoğlu et al., 2017*; *Greve, Grecco & Santos-Silva, 2009*; *Konjen, Napnark & Janchai, 2015*; *Ulusoy, Cerrahoglu & Orguc, 2017*). The study by *Ulusoy, Cerrahoglu & Orguc (2017)* found that the therapeutic US reduced pain intensity in patients with plantar fasciitis at 25% of treatment success. Their main criteria of successful treatment efficacy was a 60% or more reduction in heel pain for two or more measurements of VAS. *Akinoğlu et al. (2017)* found that pain score was reduced in the US group with no significant difference compared to other groups in the study. However, they did not mention any prespecified criteria for significant pain reduction. *Konjen, Napnark & Janchai (2015)* stated that US therapy effectively reduced pain in plantar fasciitis patients. Their criteria for successful therapy was pain less than or equal to 30 mm on the VAS measurement. *Greve, Grecco & Santos-Silva (2009)* concluded that US therapy reduced pain scores and was more effective than radial extracorporeal shock wave therapy (r-ESWT). Since all their subject had pain equal to or above five on VAS measurement before US therapy, the authors considered that US is effective in reducing pain when all subjects reported pain less than five on VAS after treatment. Only one study out of five reported that both active US and sham US relieved pain and their criteria for significant clinical difference was a 30% pain reduction of two points decrease in NPRS score (*Katzap et al., 2018*).

### Effect of ultrasound therapy on functional disability

The functional impairment of patients with plantar fasciitis was reduced by 25% after therapeutic ultrasound therapy, as *Ulusoy, Cerrahoglu & Orguc (2017)* reported (*Ulusoy, Cerrahoglu & Orguc, 2017*). *Akinoğlu et al. (2017)* Stated that the therapeutic US improved foot function through The Foot Function Index (FFI) score better than other used treatments such as r-ESWT and home exercise programs. *Konjen, Napnark & Janchai (2015)* found that US therapy improved mobility in patients with plantar fasciitis. *Greve, Grecco & Santos-Silva (2009)* reported that the therapeutic US increased the function ability in those patients with plantar fasciitis. *Katzap et al. (2018)* found that US therapy had the same effect on functional disability as a sham US.

### Effect of ultrasound therapy on the other outcome measures

Two studies found that the therapeutic US was useful in reducing tenderness and thickness of the plantar fascia in patients with plantar fasciitis (*Greve, Grecco & Santos-Silva, 2009*; *Ulusoy, Cerrahoglu & Orguc, 2017*). Moreover, according to *Akinoğlu et al. (2017)*, US therapy enhanced patients' static and dynamic balance. Regarding the periodicity of pain during the week and the number of hours of pain per day, therapeutic US was more effective in these two outcome measures.

### Characteristics of ultrasound therapy

The US therapy featured by frequency, intensity, mode and duration of application, which varied in the selected studies. *Katzap et al. (2018)* used the US with frequency 1 MHz, intensity 1.8 $W/cm^2$ and continuous mode for 8 min. However, the study by *Ulusoy,*

*Cerrahoglu & Orguc (2017)* used different parameters (frequency 1 MHz, intensity 2 W/cm$^2$, continuous mode and the session duration of 5 min). *Akinoğlu et al. (2017)* used the therapeutic US with frequency 3 MHz, Intensity 1 W/cm$^2$, a pulsed mode for 8 min. In the study conducted by *Konjen, Napnark & Janchai (2015)*, the parameters used were the same as that of *Akinoğlu et al. (2017)*, except the mode was continuous and time 10 min. The last study mentioned only the frequency (1 MHz) and Intensity (1.2 W/cm$^2$) (*Greve, Grecco & Santos-Silva, 2009*).

## DISCUSSION

This systematic review aimed to investigate the effectiveness of the therapeutic ultrasound in decreasing pain intensity and improving functional disability in patients with plantar fasciitis. Five randomised controlled studies were included in this review, of which four showed beneficial effects of therapeutic ultrasound on pain reduction and disability improvement. Our findings confirm the results of previous research in which the use of ultrasound therapy was associated with reduced pain by 24% in patients with heel pain (*Ulusoy, Cerrahoglu & Orguc, 2017*). Also, it was found that ultrasound therapy plus therapeutic infrared was effective in decreasing pain when compared with therapeutic infrared alone for the treatment of calcaneal spur (*Aydog et al., 1996*). In contrast, *Zanon, Brasil & Imamura (2006)* and *Katzap et al. (2018)* found that ultrasound therapy was not more effective than a placebo in relieving pain and functional disability in the management of plantar fasciitis. However, the variability of dosage and mode of application (continuous *vs* pulsed) might affect the level of effectiveness of the therapy.

The mechanism of the therapeutic ultrasound is believed to be the reason behind its positive effects, as it can increase the temperature and metabolism of tissues in addition to the increase of blood flow. It also helps to improve tissues' flexibility and mobility by softening them. Moreover, it increases the chemical activity and the cell membrane's permeability (*Baker, Robertson & Duck, 2001*). Additionally, it has been suggested that ultrasonic energy influences the chemical activity of tissues by making cell membranes more permeable, controlling the formation of molecules and proteins, and possibly promoting tissue recovery and speeding up the healing process (*Al-Siyabi et al., 2022*).

Furthermore, plantar fascia thickness results from the inflammatory process in the plantar fascia, makes walking painful and interferes with daily activities for the patient. The US treatment boosts cellular activity levels and circulation and reduces inflammation while having an analgesic impact with its thermal, nonthermal, mechanical, and micro-massage actions (*Akinoğlu et al., 2017*). According to a randomised controlled trial, ultrasound therapy effectively reduced plantar fascia thickness from 4.76 to 3.99 mm (*Ulusoy, Cerrahoglu & Orguc, 2017*).

Moreover, it has been found that ultrasound therapy can help in reducing soft tissue oedema and bone marrow oedema, which are common findings through magnetic resonance imaging (MRI) in individuals with plantar fasciitis (*Ulusoy, Cerrahoglu & Orguc, 2017*). Therefore, *Wong et al. (2007)* reported that most musculoskeletal physical therapists use ultrasound therapy to reduce soft tissue inflammation and pain, for improving tissue extensibility, scare tissue remodelling and soft tissue injury healing.

Ultrasound therapy can provide internal tissue massage action for plantar fascia that is produced by longitudinal waves, causing mechanical vibration bundles to vibrate, causing changes in intracellular pressure (*Krukowska et al., 2016*). These mechanical vibration bundles cause a range of regulatory phenomena that significantly reduce pain and swelling, speed up the healing process, and standardise immune responses. These phenomena result from enhanced tissue perfusion and oxygenation, quicker prosthetic group enzyme activity, the release of mediators, and enhanced cell and intercellular membrane penetrability (*Krukowska et al., 2016*).

The authors used the PRISMA protocol when performing this extensive review. The selected studies were evaluated using the PEDro scale by two reviewers. Three studies were rated highly on the PEDro scale, while two were rated at the low to moderate quality level. Because of how the intervention is structured, blinding neither subjects nor researchers was possible in four studies, which may have introduced some bias into the outcomes. Furthermore, none of the included studies tried to state their intention to treat missing data explicitly.

Even though our systematic review reported the beneficial effects of ultrasound therapy in treating plantar fasciitis, we noticed a lack of consensus on the optimal parameter for treating this condition. This may be attributed to several factors. The selected studies used different parameters in terms of frequency, intensity, mode and duration of the treatment protocol, which makes it difficult to compare. There was variability in patient characteristics, degree of pain and inflammation and duration of the symptoms. In addition, out of five selected studies, only two studies reported the duration of the symptoms. The diagnostic criteria were also varied between the studies. A standardised treatment protocol and guidelines for ultrasound therapy for treating plantar fasciitis are essential to ensure safe, effective and evidence-based care.

Several limitations were deduced in this review. First, few studies were selected in this review which might be because of the selection of the specific outcomes. However, these outcomes were the most used in the literature for people with plantar fasciitis. Second, there were variations in dosages, number of sessions, and duration of application of ultrasound therapy in the selected studies, which made the meta-analysis impossible. Further randomized control trials are recommended on this topic. Third, the search was limited to the studies published in English, and there might be other studies in different languages that can be used to draw a better conclusion about the topic. Fourth, the reviewed studies had differences in the dosages, number of sessions, and application duration, making meta-analysis of findings impossible. Fifth, only one out of five studies included a real control group, meaning no treatment or sham treatment. It would be much better if the researchers included a real control group. However, it was reported that creating a real control group without intervention is unethical (*Gupta & Verma, 2013*). We used only three databases to search for relevant studies. Among these three databases, only Pubmed is regarded as a larger database. In addition, the electronic search was conducted in October 2022, and papers published after that date were not included in this review.

### Clinical implications

This systematic review found that ultrasound therapy reduced pain and functional impairment in people with plantar fasciitis. We recommend that physical therapists can use ultrasound therapy to treat plantar fasciitis patients to reduce pain and improve function.

## CONCLUSION

Ultrasound therapy with multiple intensity and duration has shown the potential for conservative treatment of plantar fasciitis. The findings of this review confirm the efficacy of therapeutic ultrasound to treat plantar fasciitis and discuss the variety of therapy protocols. However, there is still a need future high-quality RCTs that can lead to a more conclusive analysis of the effectiveness of ultrasound therapy in cases of plantar fasciitis. It is also necessary to determine which dosages, number of sessions, and duration of application of ultrasound therapy can be most effective in treating plantar fasciitis.

### Funding

The authors received no funding for this work.

### Competing Interests

The authors declare that they have no competing interests.

### Author Contributions

- Anas Mohammed Alhakami conceived and designed the experiments, performed the experiments, prepared figures and/or tables, authored or reviewed drafts of the article, and approved the final draft.
- Reem Abdullah Babkair analyzed the data, authored or reviewed drafts of the article, and approved the final draft.
- Ahmad Sahely analyzed the data, authored or reviewed drafts of the article, and approved the final draft.
- Shibili Nuhmani conceived and designed the experiments, prepared figures and/or tables, and approved the final draft.

### Data Availability

This is a systematic review.

### Supplemental Information

Supplemental information for this article can be found online at http://dx.doi.org/10.7717/peerj.17147#supplemental-information.

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
