# Peer review of "Effectiveness of therapeutic ultrasound on reducing pain intensity and functional disability in patients with plantar fasciitis: a systematic review of randomised controlled trials"

_PeerJ, doi:10.7717/peerj.17147_

## Round 0.1 · original submission · Minor Revisions

- Please read the reviewers' comments carefully and make changes appropriately.
- In case you decide not to proceed with a change, please justify adequately the reason for this decision.
- There are some details described by the reviewers that need to be addressed.

Reviewer 1 ·

Basic reporting

Reporting of systematic review using PRISMA reporting format is commendable. However, the current review did not report on the total number of screened studies, excluded studies and reason for exclusion.

Authors should use PRISMA flowchart for adequate reporting. Follow the link to substantiate the data reporting.

https://www.bmj.com/content/372/bmj.n71

Experimental design

The study falls within the scope of the journal and answers a relevant clinical question to make informed decisions in clinical practice.

Methodology should comprehensively report the procedure from screening to selection of the included studies for review.

Validity of the findings

The supplemental data is adequate however, conclusion may be drawn by including literature gaps and how the gap has been filled by this review.

Additional comments

The authors are appreciated for reviewing a clinically relevant question, however, the reporting for the review needs more detailing.

Annotated reviews are not available for download in order to protect the identity of reviewers who chose to remain anonymous.

Reviewer 2 ·

Basic reporting

No comment.

Experimental design

No comment

Validity of the findings

The objectives of the systematic review did not include finding out the optimal parameters for the use of therapeutic utrasound in patients with plantar fasciitis. It would be very useful to know which parameters, in terms of frequency, intensity, mode and time are the most effective in reducing pain and improving function in patients with plantar fasciitis.

In the discussion, there is a scarce analysis, among the different studies, of the effect of terepeutic utrasound in patients with plantar fasciitis, especially considering the different outcome measures used and the different parameters applied. Consequently, it is not clear what would be the best way to apply ultrasound to reduce pain and improve function in patients with plantar fasciitis.

Only two of the studies reviewed indicate when the post-intervention measurement was performed. Therefore, it cannot be generalized that therapeutic ultrasound reduces plantar fasciitis pain, as it is not clear for how long the pain is reduced. Moreover, in the studies where it is specified when the post-intervention variables were measured, therapeutic ultrasound was combined with another therapeutic technique such as stretching. Thus, we do not know whether the positive effects are due to ultrasound or to stretching.

It would be advisable to provide more detail on each of the outcome measures used in the 5 studies analyzed. Especially those referring to pain and functional disability. It is important to determine when pain is measured. Indicate whether it is pain at the first step of the day, whether it is pain at the end of the day or whether it is pain throughout the day. It is also important to indicate what was the mean in the analysis of the differences in results. The conclusions indicate that the patients improve, but we cannot observe anywhere whether the subjects have improved 3 points on the VAS scale or have improved 6 points. I believe that these aspects should be detailed in the review.

Additional comments

Today we know that the ultrasound diagnosis of plantar fasciitis is very important. Therefore, it would be interesting to know what has been the method of diagnosis of plantar fasciitis. It is likely that in the studies analyzed, the participants suffered from plantar heel pain similar to plantar fasciitis pain, but that it could be a pain caused for example by atrophy of the plantar fat. I believe this is important when designing the inclusion and exclusion criteria for the review.

On the other hand, it would be advisable to include studies that provide information on how long the patient has been suffering the symptoms. It is important to differentiate between acute plantar fasciitis, of a few weeks of evolution, and chronic plantar fasciitis of even several years of evolution.

I believe that the RCTs analyzed in the systematic review should not be used to discuss the results of the study. Other different studies should be used to compare the results of the review in order to provide greater relevance to the results.

---

## Round 0.2 · Minor Revisions

Authors should add a limitation related to the databases used. Only one of them, MEDLINE (via PubMed) is large enough for a systematic review of these characteristics.

An important limitation should also be added related to the time of the bibliographic search (October 2022) or, failing that, the authors should carry out a new search to find out if anything new has been published since that date.

Reviewer 2 ·

Basic reporting

No comment

Experimental design

No comment

Validity of the findings

No comment

Additional comments

After reviewing the article for the second time, the changes made are appreciated. In this way the reader will be able to obtain a more accurate conclusion, knowing the limits of the systematic review and understanding that more research is needed to specify which is the best methodology to apply therapeutic ultrasound.

I appreciate the changes made regarding the method used for the diagnosis of plantar fasciitis, the time of measurement of the primary outcome measures as well as the changes made regarding the articles used for the discussion of the article.

---

## Round 0.3 · accepted · Accept

I confirm that the authors have addressed all of my comments.
Thank you!